# A PAINLESS ATTENTION MECHANISM FOR CONVOLUTIONAL NEURAL NETWORKS

## ABSTRACT

We propose a novel attention mechanism to enhance Convolutional Neural Networks for fine-grained recognition. The proposed mechanism reuses CNN feature activations to find the most informative parts of the image at different depths with the help of gating mechanisms and without part annotations. Thus, it can be used to augment any layer of a CNN to extract low- and high-level local information to be more discriminative.

Differently, from other approaches, the mechanism we propose just needs a single pass through the input and it can be trained end-to-end through SGD. As a consequence, the proposed mechanism is modular, architecture-independent, easy to implement, and faster than iterative approaches.

Experiments show that, when augmented with our approach, Wide Residual Networks systematically achieve superior performance on each of five different fine-grained recognition datasets: the Adience age and gender recognition benchmark, Caltech-UCSD Birds-200-2011, Stanford Dogs, Stanford Cars, and UEC Food-100, obtaining competitive and state-of-the-art scores.

## 1 INTRODUCTION

Humans and animals process vasts amounts of information with limited computational resources thanks to attention mechanisms which allow them to focus resources on the most informative chunks of information. These biological mechanisms have been extensively studied (see Anderson (1985); Desimone & Duncan (1995)), concretely those mechanisms concerning visual attention, e.g. the work done by Ungerleider & G (2000).

In this work, we inspire on the advantages of visual and biological attention mechanisms for fine-grained visual recognition with Convolutional Neural Networks (CNN) (see LeCun et al. (1998)). This is a particularly difficult task since it involves looking for details in large amounts of data (images) while remaining robust to deformation and clutter. In this sense, different attention mechanisms for fine-grained recognition exist in the literature: (i) iterative methods that process images using "glimpses" with recurrent neural networks (RNN) or long short-term memory (LSTM) (e.g. the work done by Sermanet et al. (2015); Zhao et al. (2017b)), (ii) feed-forward attention mechanisms that augment vanilla CNNs, such as the Spatial Transformer Networks (STN) by Jaderberg et al. (2015), or a top-down feed-forward attention mechanism (FAM) (Rodríguez et al. (2017)). Although it is not applied to fine-grained recognition, the Residual Attention introduced by Wang et al. (2017) is another example of feed-forward attention mechanism that takes advantage of residual connections (He et al. (2016)) to enhance or dampen certain regions of the feature maps in an incremental manner.

Inspired by all the previous research about attention mechanisms in computer vision, we propose a novel feed-forward attention architecture (see Figure 1) that accumulates and enhances most of the desirable properties from previous approaches:

1. Detect and process in detail the most informative parts of an image: more robust to deformation and clutter.

2. Feed-forward trainable with SGD: faster inference than iterative models, faster convergence rate than Reinforcement Learning-based (RL) methods like the ones presented by Sermanet et al. (2015); Liu et al. (2016).

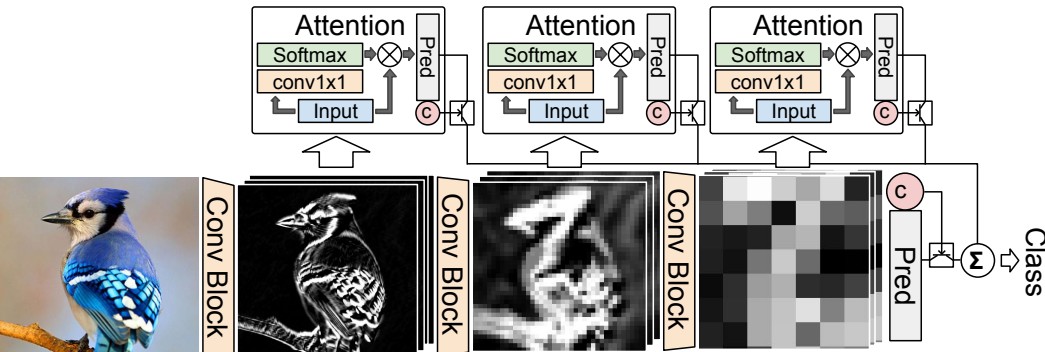

Figure 1: The proposed mechanism. Feature maps at different levels are processed to generate spatial attention masks and use them to output a class hypothesis based on local information and a confidence score (C). The final prediction consists of the average of all the hypotheses weighted by the normalized confidence scores.

3. Preserve low-level detail: unlike Residual Attention (Wang et al. (2017)), where low-level features are subject to noise after traversing multiple residual connections, our architecture directly uses them to make predictions. This is important for fine-grained recognition, where low-level patterns such as textures can help to distinguish two similar classes.

Moreover, the proposed mechanism possesses other interesting properties such as:

1. **Modular and incremental**: the attention mechanism can be replicated at each layer on any convolutional architecture, and it is easy to adapt to the task at hand.

2. **Architecture independent**: the mechanism can accept any pre-trained architecture such as VGG (Simonyan & Zisserman (2014)) or ResNet.

3. **Low computational impact**: While STNs use a small convnet to predict affine-transform parameters and Residual Attention uses the hourglass architecture, our attention mechanism consists of a single $1 \times 1$ convolution and a small fully-connected layer.

4. **Simple**: the proposed mechanism can be implemented in few lines of code, making it appealing to be used in future work.

The proposed attention mechanism has been included in a strong baseline like Wide Residual Networks (WRN) (Zagoruyko & Komodakis (2016)), and applied on five fine-grained recognition datasets. The resulting network, called Wide Residual Network with Attention (WRNA) systematically enhances the performance of WRNs, obtains competitive results using low resolution training images, and surpasses the state of the art in the Adience gender recognition task, Stanford dogs, and UEC Food-100. Table 1 shows the gain in performance of WRNA w.r.t. WRN for all the datasets considered in this paper.

In the next section, we review the most relevant work concerning attention mechanisms for visual fine-grained recognition.

## 2 RELATED WORK

As reviewed by Zhao et al. (2017a), there are different approaches to fine-grained recognition: (i) vanilla deep CNNs, (ii) CNNs as feature extractors for localizing parts and do alignment, (iii) ensembles, (iv) attention mechanisms.

In this paper we focus on (iv), the attention mechanisms, which aim to discover the most discriminative parts of an image to be processed in greater detail, thus ignoring clutter and focusing on the most distinctive traits. These parts are central for fine-grained recognition, where the inter-class variance is small and the intra-class variance is high.

| Task | WRN | WRNA | Δ |
|------|-----|------|------|
| Age | 58.6 | 59.7 | **+1.1** |
| Birds | 81.0 | 82.0 | **+1.0** |
| Food | 84.3 | **85.5** | **+1.2** |
| Dogs | 89.6 | **89.9** | **+0.4** |
| Cars | 87.8 | 90.0 | **+2.2** |
| Gend | 93.9 | **94.8** | **+0.9** |

Table 1: Performance of WRN and our approach (WRA) on the Adience benchmark for Age, Gender (Gend); CUB200-2011 (Birds); Stanford cars (Cars); Stanford dogs (Dogs); and UEC Food-100 (Food). The absolute accuracy improvement (in %) is marked as Δ. Bold performance indicates outperforming the state of the art. The augmented network consistently outperforms the baseline up to a relative 18% relative error decrease on Cars.

Different fine-grained attention mechanisms can be found in the literature. Xiao et al. (2015) proposed a *two-level attention* mechanism for fine-grained classification on different subsets of the ICLR2012 (Russakovsky et al. (2012)) dataset, and the CUB200_2011. In this model, images are first processed by a bottom-up object proposal network based on R-CNN (Zhang et al. (2014)) and selective search (Uijlings et al. (2013)). Then, the softmax scores of another ILSVRC2012 pre-trained CNN, which they call *FilterNet*, are thresholded to prune the patches with the lowest parent class score. These patches are then classified to fine-grained categories with a *DomainNet*. Spectral clustering is also used on the *DomainNet* filters in order to extract parts (head, neck, body, etc.), which are classified with an SVM. Finally, the part- and object-based classifier scores are merged to get the final prediction. The *two-level attention* obtained state of the art results on CUB200-2011 with only class-level supervision. However, the pipeline must be carefully fine-tuned since many stages are involved with many hyper-parameters.

Differently from *two-level attention*, which consists of independent processing and it is not end-to-end, Sermanet *et al.* proposed to use a deep CNN and a Recurrent Neural Network (RNN) to accumulate high multi-resolution "glimpses" of an image to make a final prediction (Sermanet et al. (2015)), however, reinforcement learning slows down convergence and the RNN adds extra computation steps and parameters.

A more efficient approach was presented by Liu *et al.* in (Liu et al. (2016)), where a fully-convolutional network is trained with reinforcement learning to generate confidence maps on the image and use them to extract the parts for the final classifiers whose scores are averaged. Compared to previous approaches, in the work done by Liu et al. (2016), multiple image regions are proposed in a single timestep thus, speeding up the computation. A greedy reward strategy is also proposed in order to increase the training speed. The recent approach presented by Fu et al. (2017) uses a classification network and a recurrent attention proposal network that iteratively refines the center and scale of the input (RA-CNN). A ranking loss is used to enforce incremental performance at each iteration.

Zhao *et al.* proposed *Diversified Visual Attention Network* (DVAN), *i.e.* enforcing multiple non-overlapped attention regions (Zhao et al. (2017b)). The overall architecture consists of an attention canvas generator, which extracts patches of different regions and scales from the original image; a VGG-16 (Simonyan & Zisserman (2014)) CNN is then used to extract features from the patches, which are aggregated with a DVAN long short-term memory (Hochreiter & Schmidhuber (1997)) that attends to non-overlapping regions of the patches. Classification is performed with the average prediction of the DVAN at each region.

All the previously described methods involve multi-stage pipelines and most of them are trained using reinforcement learning (which requires sampling and makes them slow to train). In contrast, STNs, FAM, and our approach jointly propose the attention regions and classify them in a single pass. Moreover, they possess interesting properties compared to previous approaches such as (i) simplicity (just a single model is needed), (ii) deterministic training (no RL), and (iii) feed-forward training (only one timestep is needed), see Table 2. In addition, since our approach only uses one CNN stream, it is far more computationally efficient than STNs and FAM, as described next.

| Publication | Single Stream | Single Pass | SGD Trainable |
|---|---|---|---|
| Sermanet et al. (2015) | × | × | × |
| Xiao et al. (2015) | × | × | × |
| Liu et al. (2016) | × | × | × |
| Fu et al. (2017) | × | × | ✓ |
| Rodríguez et al. (2017) | × | × | ✓ |
| Lin et al. (2017) | × | × | ✓ |
| Jaderberg et al. (2015) | ×* | ✓ | ✓ |
| Zhao et al. (2017b) | ✓ | × | ✓ |
| **Ours** | ✓ | ✓ | ✓ |

Table 2: Comparison of Attention models in the Computer Vision Literature. *Single Stream*: input data is fed through a single CNN tower. *Single pass*: train and inference outputs are obtained in a single pass trough the model. *SGD Trainable*: the model can be trained end-to-end with SGD. *Multiple Regions*: the attention mechanism can extract information of multiple regions of the image at once. (*) the size of the attention module is unbounded (thus could consist of a whole CNN pipeline).

## 3 OUR APPROACH

Our approach consists of a universal attention module that can be added after each convolutional layer without altering pre-defined information pathways of any architecture. This is helpful since it allows to seamlessly augment any architecture such as VGG and ResNet with no extra supervision, *i.e.* no part labels are necessary. The attention module consists of three main submodules: (i) the attention heads $\mathcal{H}$, which define the most relevant regions of a feature map, (ii) the output heads $\mathcal{O}$, generate an hypothesis given the attended information, and (iii) the confidence gates $\mathcal{G}$, which output a confidence score for each attention head. Each of these modules is explained in detail in the following subsections.

### 3.1 OVERVIEW

As it can be seen in Fig 1, 2a, and 2b, a $1 \times 1$ convolution is applied to the output of the augmented layer, producing an attentional heatmap. This heatmap is then element-wise multiplied with a copy of the layer output, and the result is used to predict the class probabilities and a confidence score. This process is applied to an arbitrary number $N$ of layers, producing $N$ class probability vectors, and $N$ confidence scores. Then, all the class predictions are weighted by the confidence scores (softmax normalized so that they add up to 1) and averaged (using 9). This is the final combined prediction of the network.

### 3.2 ATTENTION HEAD

Inspired by DVAN (Zhao et al. (2017b)) and the *transformer* architecture presented by Vaswani et al. (2017), and following the notation established by Zagoruyko & Komodakis (2016), we have identified two main dimensions to define attentional mechanisms: (i) the number of layers using the attention mechanism, which we call *attention depth* (AD), and (ii) the number of attention heads in each attention module, which we call *attention width* (AW). Thus, a desirable property for any universal attention mechanism is to be able to be deployed at any arbitrary *depth* and *width*.

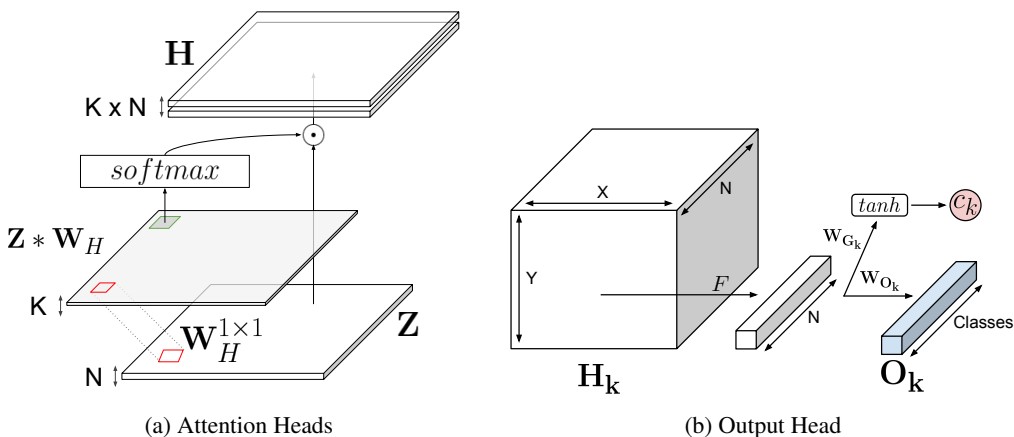

(a) Attention Heads

(b) Output Head

Figure 2: Scheme of the submodules in the proposed mechanism. (a) depicts the attention heads, (b) shows a single output head.

This property is fulfilled by including $K$ attention heads $\mathcal{H}_k$ (width), depicted in Figure 2a, into each attention module (depth)[1]. Then, the attention heads at layer $l$, receive the feature activations $\mathbf{Z}^l$ of that layer as input, and output $K$ weighted feature maps, see Equations 1 and 2:

$$\mathbf{M}^l = spatial\_softmax(\mathbf{W}_H^l * \mathbf{Z}^l), \tag{1}$$

$$\mathbf{H}^l = \mathbf{M}^l \odot \mathbf{Z}^l, \tag{2}$$

where $\mathbf{H}^l$ is the output matrix of the $l^{th}$ attention module, $\mathbf{W}_H$ is a $1 \times 1$ convolution kernel with output dimensionality $K$ used to compute the attention masks corresponding to the attention heads $\mathbf{H}_k$, $*$ denotes the convolution operator, and $\odot$ is the element-wise product. Please note that $\mathbf{M}_k^l$ is a 2d flat mask and the product with each of the $N$ input channels of $\mathbf{Z}$ is done by broadcasting. Likewise, the dimensionality of $\mathbf{H}_k^l$ is the same as $\mathbf{Z}^l$. The spatial softmax is used to enforce the model to learn the most relevant region of the image. Sigmoid units could also be used at the risk of degeneration to all-zeros or all-ones.

Since the different attention heads in an attention module are sometimes focusing on the same exact part of the image, similarly to DVAN, we have introduced a regularization loss $L_R$ that forces the multiple masks to be different. In order to simplify the notation, we set $\mathbf{m}_k^l$ to be $k^{th}$ flattened version of the attention mask $\mathbf{M}$ in Equation 1. Then, the regularization loss is expressed as:

$$L_R = \sum_{i=1}^{K} \sum_{j \neq i} ||\mathbf{M}_i^l (\mathbf{M}_j^l)^T||_2^2, \tag{3}$$

*i.e.*, it minimizes the squared Frobenius norm of the off-diagonal cross-correlation matrix formed by the squared inner product of each pair of different attention masks, pushing them towards orthogonality ($L_R = 0$). This loss is added to the network loss $L_{net}$ weighted by a constant factor $\gamma = 0.1$ which was found to work best across all tasks:

$$L_{net}^* = L_{net} + \gamma L_R \tag{4}$$

---

[1]Notation: $\mathcal{H}, \mathcal{O}, \mathcal{G}$ are the set of attention heads, output heads, and attention gates respectively. Uppercase letters are used as functions or constants, and lowercase letters are used as indices. Bold uppercase represent matrices and bold lowercase represent vectors.

| AD | AW | G | $\Delta$ |
|----|----|---|------|
| 1 | 1 |  | +1.2 |
| 2 | 1 |  | +1.4 |
| 2 | 2 |  | +1.5 |
| 2 | 2 | ✓ | +1.6 |

Table 3: Average performance impact across datasets on (in accuracy %) of the attention depth ($AD$), attention width ($AW$), and the presence of gates ($G$) on WRN.

### 3.3 OUTPUT HEAD

The output of each attention module consists of a spatial dimensionality reduction layer:

$$F : \mathbb{R}^{x \times y \times n} \to \mathbb{R}^{1 \times n}, \tag{5}$$

followed by a fully-connected layer that produces an hypothesis on the output space, see Figure 2b.

$$\mathbf{o}^l = F(\mathbf{H}^l)\mathbf{W}_O^l \tag{6}$$

We consider two different dimensionality reductions: (i) a channel-wise inner product by $\mathbf{W}_F^{1 \times n}$, where $\mathbf{W}_F$ is a dimensionality reduction projection matrix with $n$ the number of input channels; and (ii) an average pooling layer. We empirically found (i) to work slightly better than (ii) but at a higher computational cost. $\mathbf{W}_F$ is shared across all attention heads in an attention module.

### 3.4 ATTENTION GATES

Each attention module makes a class hypothesis given its local information. However, in some cases, the local features are not good enough to output a good hypothesis. In order to alleviate this problem, we make each attention module, as well as the network output, to predict a confidence score $\mathbf{c}$ by means of an inner product by the gate weight matrix $\mathbf{W}_G$:

$$\mathbf{c}^l = tanh(F(\mathbf{H}^l)\mathbf{W}_G^l). \tag{7}$$

The gate weights $\mathbf{g}$ are then obtained by normalizing the set of scores by means of a $softmax$ function:

$$g_k^l = \frac{e^{c_k^l}}{\sum_{i=1}^{|\mathcal{G}|} e^{c_i}}, \tag{8}$$

where $|\mathcal{G}|$ is the total number of gates, and $c_i$ is the $i^{th}$ confidence score from the set of all confidence scores. The final output of the network is the weighted sum of the output heads:

$$\mathbf{output} = g_{net} \cdot \mathbf{output}_{net} + \sum_{h\ in\{1..|\mathcal{H}|\}} \sum_{k \in \{1..K\}} g_k^l \cdot \mathbf{o}_h^l, \tag{9}$$

where $g_{net}$ is the gate value for the original network output ($\mathbf{output}_{net}$), and $\mathbf{output}$ is the final output taking the attentional predictions $\mathbf{o}_h^l$ into consideration. Please note that setting the output of $\mathcal{G}$ to $\frac{1}{|\mathcal{G}|}$, corresponds to averaging all the outputs. Likewise, setting $\{\mathcal{G} \backslash G_{output}\} = 0, G_{output} = 1$, *i.e.* the set of attention gates is set to zero and the output gate to one, corresponds to the original pre-trained model without attention.

In Table 3 we show the importance of each submodule of our proposal on WRN. Instead of augmenting all the layers of the WRN, in order to have the minimal computational impact and to attend features of different levels, attention modules are placed after each pooling layer, where the spatial resolution is divided by two. Attention Modules are thus placed starting from the fourth pooling

layer and going backward when $AD$ increases. As it can be seen, just adding a single attention module with a single attention head is enough to increase the mean accuracy by 1.2%. Adding extra heads and gates increase an extra 0.1% each. Since the first and second pooling layers have a big spatial resolution, the receptive field for $AD > 2$ was too small and did not result in increased accuracy.

The fact that the attention mask is generated by just one $1 \times 1$ convolution and the direct connection to the output makes the module fast to learn, thus being able to generate foreground masks from the beginning of the training and refining them during the following epochs. A sample of these attention masks for each dataset is shown on Figure 3.

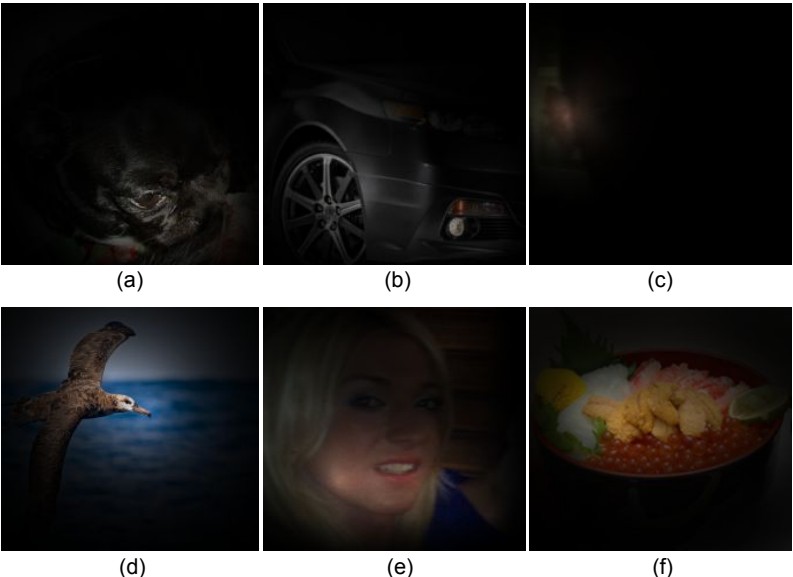

Figure 3: Attention masks for each dataset: (a) dogs, (b) cars, (c) gender, (d) birds, (e) age, (f) food. As it can be seen, the masks help to focus on the foreground object. In (c), the attention mask focuses on ears for gender recognition, possibly looking for earrings.

In the next section, we test the performance of the previously described mechanisms on different datasets.

## 4 EXPERIMENTS

In this section, we first test the design principles of our approach through a set of experiments on Cluttered Translated MNIST, and then demonstrate the effectiveness of the proposed mechanisms for fine-grained recognition on five different datasets: and age and gender (Adience dataset by Eidinger et al. (2014)), birds (CUB200-2011 by Wah et al. (2011)), Stanford cars Krause et al. (2013), Stanford dogs (Khosla et al. (2011)), and food (UECFOOD-100 by Matsuda et al. (2012)).

### 4.1 CLUTTERED TRANSLATED MNIST

In order to support the design decisions of Section 3, we follow the procedure of Mnih et al. (2014), and train a CNN on the Cluttered Translated MNIST dataset[2], consisting of $40 \times 40$ images containing a randomly placed MNIST digit and a set of $D$ randomly placed distractors, see Figure 4a. The distractors are random $8 \times 8$ patches from other MNIST digits. The CNN consists of five $3 \times 3$ convolutional layers and two fully-connected in the end, the three first convolution layers are followed by a spatial pooling. Batch-normalization was applied to the inputs of all these layers. Attention modules were placed starting from the fifth convolution (or pooling instead) backwards until $AD$

---

[2]https://github.com/deepmind/mnist-cluttered

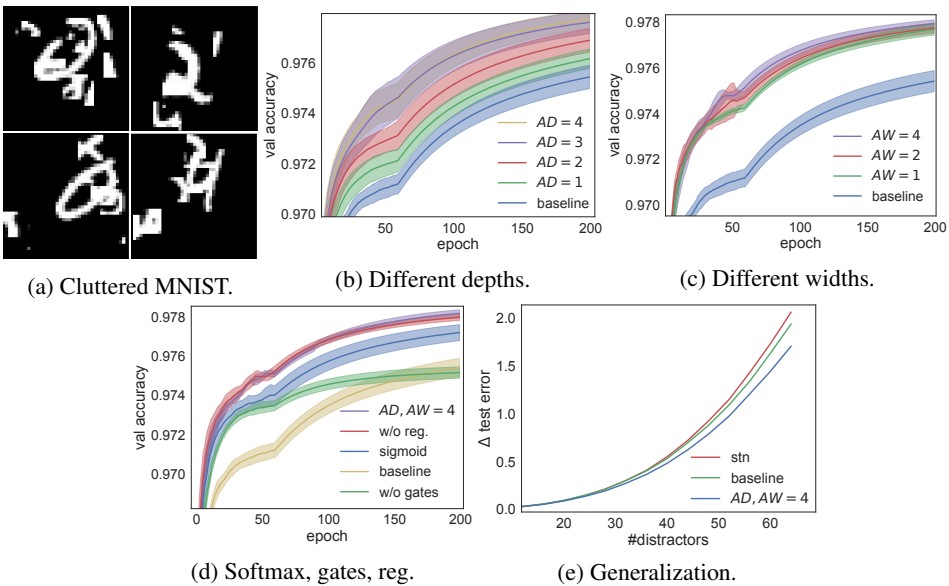

Figure 4: Ablation experiments on Cluttered Translated MNIST.

is reached. Training is performed with SGD for 200 epochs, and a learning rate of 0.1, which is divided by 10 after epoch 60. Models are trained on a $200k$ images train set, validated on a $100k$ images validation set, and tested on $100k$ test images.

First, we tested the importance of $AW$ and $AD$ for our model. As it can be seen in Figure 4b, greater $AD$ results in better accuracy, reaching saturation at $AD = 4$, note that for this value the receptive field of the attention module is $5 \times 5\ px$, and thus the performance improvement from such small regions is limited. Figure 4c shows training curves for different values of $AW$. As it can be seen, small performance increments are obtained by increasing the number of attention heads despite there is only one object present in the image.

Then, we used the best $AD$ and $AW$ to verify the importance of using softmax on the attention masks instead of sigmoid (1), the effect of using gates (Eq. 8), and the benefits of regularization (Eq. 3). Figure 4d confirms that, ordered by importance: gates, softmax, and regularization result in accuracy improvement, reaching $97.8\%$. Concretely, we found that gates pay an important role discarding the distractors, especially for high $AW$ and high $AD$.

Finally, in order to verify that attention masks are not overfitting on the data, and thus generalize to any amount of clutter, we run our best model so far (Figure 4d) on the test set with an increasing number of distractors (from 4 to 64). For the comparison, we included the baseline model before applying our approach and the same baseline augmented with an STN Jaderberg et al. (2015) that reached comparable performance as our best model in the validation set. All three models were trained with the same dataset with eight distractors. Remarkably, as it can be seen in Figure 4e, the attention augmented model demonstrates better generalization than the baseline and the STN.

## 4.2 RESULTS

In order to demonstrate that the proposed generalized attention can easily augment any recent architecture, we have trained a strong baseline, namely a Wide Residual Network (WRN) (Zagoruyko & Komodakis (2016)) pre-trained on the ImageNet. We chose to place attention modules after each pooling layer to extract different level features with minimal computational impact. The modules described in the previous sections have been implemented on pytorch, and trained in a single workstation with two NVIDIA 1080Ti. All the experiments are trained for 100 epochs, with a batch size of 64. The learning rate is first set to $10^{-3}$ to all layers except the attention modules and the classifier, for which it ten times higher. The learning rate is reduced by a factor of 0.5 every 30 iterations and the experiment is automatically stopped if a plateau is reached. The network is trained with

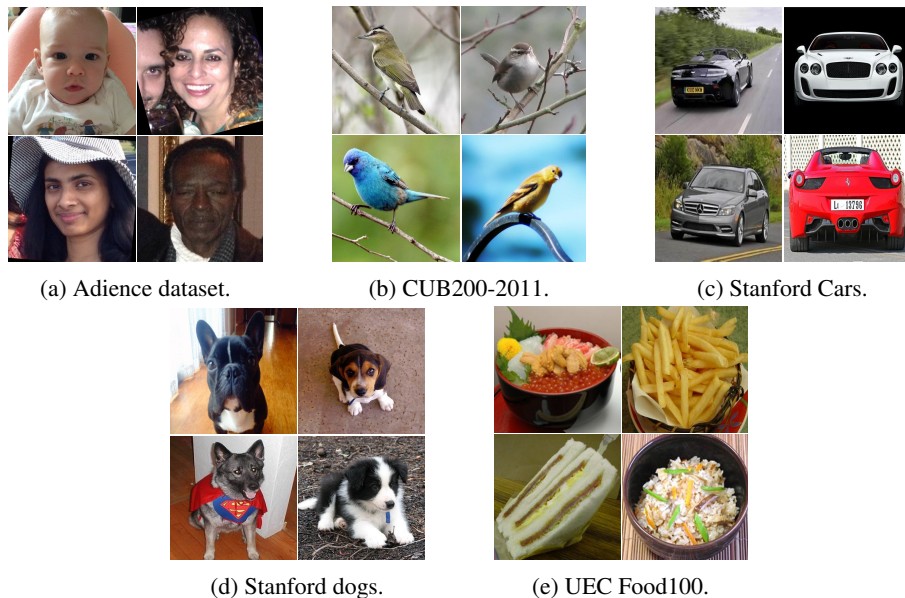

(a) Adience dataset.   (b) CUB200-2011.   (c) Stanford Cars.

(d) Stanford dogs.   (e) UEC Food100.

Figure 5: Samples from the five fine-grained datasets.

standard data augmentation, *i.e.* random $224 \times 224$ patches are extracted from $256 \times 256$ images with random horizontal flips[3]. Training curves are included in the Appendix.

For the sake of clarity and since the aim of this work is to demonstrate that the proposed mechanism universally improves CNNs for fine-grained recognition, we follow the same training procedure in all datasets. Thus, we do not use $512 \times 512$ images which are central in STNs, RA-CNNs, or B-CNNs to reach state of the art performances. Accordingly, we do not perform color jitter and other advanced augmentation techniques such as the ones used by Hassannejad et al. (2016) for food recognition. The proposed method is able to obtain state of the art results in Adience Gender, Stanford dogs and UEC Food-100 even when trained with lower resolution.

In the following subsections the proposed approach is evaluated on the five datasets.

**Adience dataset.** The adience dataset consists of 26.5 K images distributed in eight age categories (02, 46, 813, 1520, 2532, 3843, 4853, 60+), and gender labels. A sample is shown in Figure 5a.

The performance on this dataset is measured by both the accuracy in gender and age recognition tasks using 5-fold cross-validation in which the provided folds are subject-exclusive. The final score is given by the mean of the accuracies of the five folds. This dataset is particularly challenging due to the high level of deformation of face pictures taken in the wild, occlusions and clutter such as sunglasses, hats, and even multiple people in the same image.

As it can be seen in Table 4, the Wide ResNet augmented with generalized attention surpasses the baseline performance, etc.

**Caltech-UCSD Birds 200** The birds dataset (see Figure 5b) consists of 6K train and 5.8K test bird images distributed in 200 categories. The dataset is especially challenging since birds are in different poses and orientations, and correct classification often depends on texture and shape details. Although bounding box, bough segmentation, and attributes are provided, we perform raw classification as done by Jaderberg et al. (2015).

In Table 5, the performance of our approach is shown in context with the state of the art. Please note that even our approach is trained in lower resolution crops, *i.e.* $224 \times 224$ instead of $448 \times 448$, we reach the same accuracy as the recent fully convolutional attention by Liu et al. (2016).

---

[3]The code will be publicly available on github.

| Model | Publication | DSP | age | gender |
|---|---|---|---|---|
| CNN | Levi & Hassner (2015) | | 50.7 | 86.8 |
| VGG-16 | Ozbulak et al. (2016)* | ✓ | 57.9 | - |
| FAM | Rodríguez et al. (2017)* | ✓ | 61.8 | 93.0 |
| DEX | Rothe et al. (2016)** | ✓ | 64.0 | - |
| WRN | Zagoruyko & Komodakis (2016) | | 58.6 | 93.9 |
| **WRNA** | This work | | 59.7 | 94.8 |

Table 4: Performance on the adience dataset. **DSP** indicates *Domain-Specific Pre-training*, *i.e.* pre-training on millions of faces.

| Model | Publication | High Res. | Accuracy |
|---|---|---|---|
| FCAN | Liu et al. (2016) | ✓ | 82.0 |
| PD | Zhang et al. (2016) | ✓ | 82.6 |
| B-CNN | Lin et al. (2015) | ✓ | 84.1 |
| STN | Jaderberg et al. (2015) | ✓ | 84.2 |
| RA-CNN | Fu et al. (2017) | ✓ | 85.3 |
| WRN | Zagoruyko & Komodakis (2016) | | 81.0 |
| **WRNA** | This work | | 82.0 |

Table 5: Performance on Caltech-UCSD Birds 200. **High Res.** indicates whether training is performed with images with resolution higher than $224 \times 224$.

**Stanford Cars.** The Cars dataset contains 16K images of 196 classes of cars, see Figure 5c. The data is split into 8K training images and 8K testing images. The difficulty of this dataset resides in the identification of the subtle differences that distinguish between two car models.

| Model | Publication | High Res. | Accuracy |
|---|---|---|---|
| DVAN | Zhao et al. (2017b) | | 87.1 |
| FCAN | Liu et al. (2016) | ✓ | 89.1 |
| B-CNN | Lin et al. (2017) | ✓ | 91.3 |
| RA-CNN | Fu et al. (2017) | ✓ | 92.5 |
| WRN | Zagoruyko & Komodakis (2016) | | 87.8 |
| **WRNA** | This work | | 90.0 |

Table 6: Performance on Stanford Cars. **High res.** indicates that resolutions higher than $256 \times 256$ are used.

In Table 6 the performance of our approach with respect to the baseline and other state of the art is shown. The augmented WRN shows better performance than the baseline, and even surpasses recent approaches such as FCAN.

**Stanford Dogs.** The Stanford Dogs dataset consists of 20.5K images of 120 breeds of dogs, see Figure 5d. The dataset splits are fixed and they consist of 12k training images and 8.5K validation images. Pictures are taken in the wild and thus dogs are not always a centered, unique, pose-normalized object in the image but a small, cluttered region.

Table 7 shows the results on Stanford dogs. As it can be seen, performances are low in general and nonetheless, our model was able to increase the accuracy by a 0.3% (0.1% w/o gates), being the highest score obtained on this dataset to the best of our knowledge. This performance has been achieved thanks to the gates, which act as a detection mechanism, giving more importance to those attention masks that correctly guessed the position of the dog.

**UEC Food 100** is a Japanese food dataset with 14K images of 100 different dishes, see Figure 5e. Pictures present a high level of variation in the form of deformation, rotation, clutter, and noise. In

| Model | Publication | High Res. | Accuracy |
|-------|-------------|-----------|----------|
| VGG-16 | Simonyan & Zisserman (2014) | | 76.7 |
| DVAN | Zhao et al. (2017b) | | 81.5 |
| FCAN | Liu et al. (2016) | ✓ | 84.2 |
| RA-CNN | Fu et al. (2017) | ✓ | 87.3 |
| WRN | Zagoruyko & Komodakis (2016) | | 89.6 |
| **WRNA** | This work | | 89.9 |

Table 7: Performance on Stanford Dogs. **High res.** indicates that resolutions higher than $256 \times 256$ are used.

order to follow the standard procedure in the literature (*e.g.* Chen & Ngo (2016); Hassannejad et al. (2016)), we use the provided bounding boxes to crop the images before training.

| Approach | Publication | Accuracy |
|----------|-------------|----------|
| DCNN-FOOD | Yanai & Kawano (2015) | 78.8 |
| VGG | Chen & Ngo (2016) | 81.3 |
| Inception V3 | Hassannejad et al. (2016) | 81.5 |
| WRN | Zagoruyko & Komodakis (2016) | 84.3 |
| **WRNA** | This work | 85.5 |

Table 8: Performance on UEC Food-100.

Table 8 shows the performance of our model compared to the state of the art. As it can be seen, our model is able to improve the baseline by a relative 7% with a $85.5\%$ of accuracy, the best-reported result compared to previous publications.

## 5 CONCLUSION

We have presented a novel attention mechanism to improve CNNs for fine-grained recognition. The proposed mechanism finds the most informative parts of the CNN feature maps at different depth levels and combines them with a gating mechanism to update the output distribution.

Moreover, we thoroughly tested all the components of the proposed mechanism on Cluttered Translated MNIST, and demonstrate that the augmented models generalize better on the test set than their plain counterparts. We hypothesize that attention helps to discard noisy uninformative regions, avoiding the network to memorize them.

Unlike previous work, the proposed mechanism is modular, architecture independent, fast, and simple and yet WRN augmented with it show higher accuracy in each of the following tasks: Age and Gender Recognition (Adience dataset), CUB200-2011 birds, Stanford Dogs, Stanford Cars, and UEC Food-100. Moreover, state of the art performance is obtained on gender, dogs, and cars.

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

APPENDIX

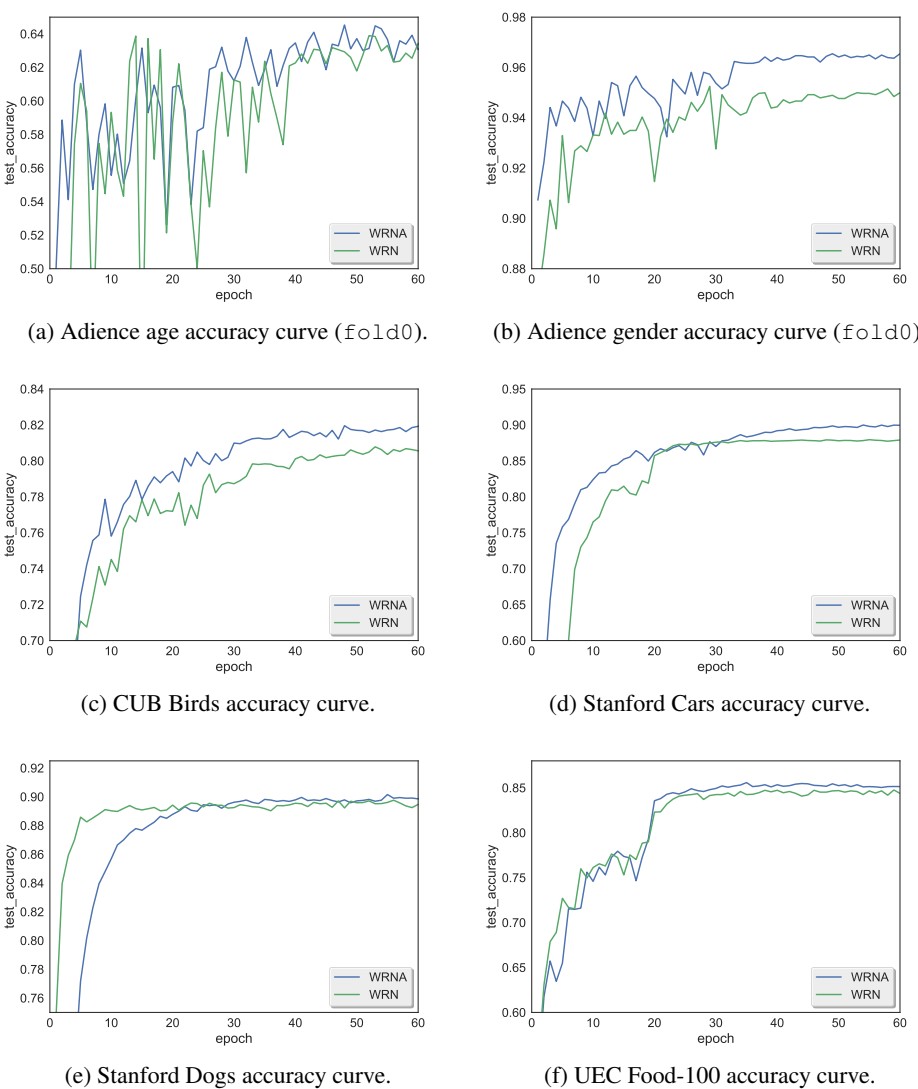

(a) Adience age accuracy curve (`fold0`).

(b) Adience gender accuracy curve (`fold0`).

(c) CUB Birds accuracy curve.

(d) Stanford Cars accuracy curve.

(e) Stanford Dogs accuracy curve.

(f) UEC Food-100 accuracy curve.

Figure 6: Test accuracy logs for the five fine-grained datasets. As it can be seen, the augmented models (WRNA) achieve higher accuracy at similar convergence rates. For the sake of space we only show one of the five folds of the Adience dataset.

