# OpenReview forum: "A Painless Attention Mechanism for Convolutional Neural Networks"
_ICLR.cc/2018/Conference — Reject_

### Official Review · AnonReviewer3 · 2017-11-27
**A limited evaluation and a limited contribution**

**Rating:** 5
**Confidence:** 4

**Review:**

The manuscript describes a novel attentional mechanism applied to fine-grained recognition.

On the positive side, the approach seems to consistently improve the recognition accuracy of the baseline  (a wide residual net). The approach is also consistently tested on the main fine-grained recognition datasets (the Adience age and gender recognition benchmark, Caltech-UCSD Birds-200-2011, Stanford Dogs, Stanford Cars, and UEC Food-100).

On the negative side, the paper could be better written and motivated.

First, some claimed are made about how the proposed approach "enhances most of the desirable properties from previous approaches” (see pp 1-2) but these claims are never backed up. More generally since the paper focuses on attention, other attentional approaches should be used as benchmarks beyond the WRN baseline. If the authors want to claim that the proposed approach is "more robust to deformation and clutter” then they should design an experiment that shows that this is the case.

Beyond, the approach seems a little ad hoc. No real rationale is provided for the different mechanisms including the gating etc and certainly no experimental validation is provided to demonstrate the need for these mechanisms. More generally, it is not clear from reading the paper specifically what computational limitation of the CNN is being solved by the proposed attentional mechanism.

Some of the masks shown in Fig 3 seem rather suspicious and prompt this referee to think that the networks are seriously overfitting to the data. For instance, why would attending to a right ear help in gender recognition?

The proposed extension adds several hyperparameters (for instance the number K of attention heads). Apologies if I missed it but I am not clear how this was optimized for the experiments reported. In general, the paper could be clearer. For instance, it is not clear from either the text or Fig 2 how H goes from XxYxK for the attention head o XxYxN for the output head.

As a final point, I would say that while some of the criticisms could be addressed in a revision, the improvements seem relatively modest. Given that the focus of the paper is already limited to fine-grained recognition, it seems that the paper would be better suited for a computer vision conference.


Minor point:

"we incorporate the advantages of visual and biological attention mechanisms” not sure this statement makes much sense. Seems like visual and biological are distinct attributes but visual attention can be biological (or not, I guess) and it is not clear how biological the proposed approach is. Certainly no attempt is made by the authors to connect to biology.

"top-down feed-forward attention mechanism” -> it should be just feed-forward attention. Not clear what "top-down feed-forward” attention could be...

---

> ### Author Response · Authors · 2018-01-01
> **Re: AnonReviewer3**
>
> >> For instance, why would attending to a right ear help in gender recognition?
>
> In the Adience dataset, most women wear earrings, so the network might have learned to look at ears whenever possible.
>
> >> The proposed extension adds several hyperparameters (for instance the number K of attention heads). Apologies if I missed it but I am not clear how this was optimized for the experiments reported. In general, the paper could be clearer. For instance, it is not clear from either the text or Fig 2 how H goes from XxYxK for the attention head o XxYxN for the output head.
>
> In Figure 2a, Z (of size XxYxN) is convolved with K XxYx1 masks. When these are multiplied by Z again, we obtain K XxYxN feature maps (broadcasting). Figure 2b depicts the output process for 1 of the K XxYxN feature maps. We have updated Figure2 to include all this information.
>
> >> As a final point, I would say that while some of the criticisms could be addressed in a revision, the improvements seem relatively modest.
>
> Given that we use a strong baseline such a WRN, we do not think that the improvements are modest. Please note that other relevant papers such as Spatial Transformer Networks [A] only reported an improvement of 0.8% on CUB200-2011 with respect to their own baseline (see table 3 in their work), and residual attention networks report 0.05% improvement on Cifar100 [B]. Most importantly, the improvement is consistently obtained across datasets, and in 3 datatsets we outperform current state of the art.
>
> [A] Jaderberg, M., Simonyan, K., & Zisserman, A. (2015). Spatial transformer networks. In Advances in Neural Information Processing Systems (pp. 2017-2025).
> [B] Wang, F., Jiang, M., Qian, C., Yang, S., Li, C., Zhang, H., ... & Tang, X. (2017). Residual Attention Network for Image Classification. CVPR2017.
>
> >> Given that the focus of the paper is already limited to fine-grained recognition, it seems that the paper would be better suited for a computer vision conference.
>
> This work helps to build better feature representations applied to Computer Vision, which it is clearly inside the scope of this conference, from the website (http://www.iclr.cc/): “The performance of machine learning methods is heavily dependent on the choice of data representation (or features) on which they are applied…. Applications in vision, audio, speech, natural language processing, robotics, neuroscience, or any other field...”
>
> >>Minor point:  "we incorporate the advantages of visual and biological attention mechanisms” not sure this statement makes much sense. Seems like visual and biological are distinct attributes but visual attention can be biological (or not, I guess) and it is not clear how biological the proposed approach is. Certainly no attempt is made by the authors to connect to biology.
>
> The way the proposed mechanism relates to biological attention is similar to the relationship between artificial and real neural networks, this is similarly done in [A]. Thus we have corrected the statement for “we incorporate the advantages inspired by visual and biological attention mechanisms, as stated in [A]”
>
> [A] Ba, J., Mnih, V., & Kavukcuoglu, K. (2014). Multiple object recognition with visual attention. ICLR2015.
>
> >> "top-down feed-forward attention mechanism” -> it should be just feed-forward attention. Not clear what "top-down feed-forward” attention could be…
>
> In the literature, bottom-up attention is referred to the process of finding the most relevant regions of the image at the feature level, i.e. regions that are salient from their surroundings, while top-down attention refers to a high level process which finds the most relevant part of an input taking into account global information [A] (in a CNN top-down usually means to choose the regions to attend at the output instead of directly doing it at the feature level, which is the case for [B,C].)
>
> [A] Connor, Charles E., Howard E. Egeth, and Steven Yantis. "Visual attention: bottom-up versus top-down." Current Biology14.19 (2004): R850-R852.
> [B] Oliva, A., Torralba, A., Castelhano, M. S., & Henderson, J. M. (2003, September). Top-down control of visual attention in object detection. In Image processing, 2003. icip 2003. proceedings. 2003 international conference on (Vol. 1, pp. I-253). IEEE.
> [C] Rodríguez, P., Cucurull, G., Gonfaus, J. M., Roca, F. X., & Gonzalez, J. (2017). Age and gender recognition in the wild with deep attention. Pattern Recognition, 72, 563-571.

---

> ### Author Response · Authors · 2018-01-01
> **Re: AnonReviewer3**
>
> >> If the authors want to claim that the proposed approach is "more robust to deformation and clutter” then they should design an experiment that shows that this is the case.
>
> In the new introduced experiments on Cluttered Translated MNIST (Section 4.1 in the new version of the paper), we confirm that indeed the proposed method is more robust than the baseline.
>
> >> Beyond, the approach seems a little ad hoc. No real rationale is provided for the different mechanisms including the gating etc and certainly no experimental validation is provided to demonstrate the need for these mechanisms.
>
> In section 3, the rationale for the different mechanisms is mentioned in each of the subsections. For instance, gates regulate the relative importance of the predictions of each attention head. This is important when AW is high and the current input has a few informative regions. In this case, just one attention head would be enough and thus, heads focusing in other regions can be dampened by the gates. This explanation is now added in section 3.4 and the conclusion.
> In addition, we have included experiments on cluttered MNIST showing that gates are critical to obtaining good performances with high AW (see Figure 4d in the new version of the paper).
>
> >> More generally, it is not clear from reading the paper specifically what computational limitation of the CNN is being solved by the proposed attentional mechanism.
>
> The proposed paper addresses the same problem as other attentional methods in the literature [A,B,C, ...], i.e. it enhances the model to find the most informative parts of the image and to discard irrelevant information. This is especially relevant for fine-grained recognition, where some details are more informative than other salient features of the image.
>
> [A] Ba, J., Mnih, V., & Kavukcuoglu, K. (2014). Multiple object recognition with visual attention. ICLR2015.
> [B] Xu, K., Ba, J., Kiros, R., Cho, K., Courville, A., Salakhudinov, R., ... & Bengio, Y. (2015, June). Show, attend and tell: Neural image caption generation with visual attention. In International Conference on Machine Learning (pp. 2048-2057).
> [C] Jaderberg, M., Simonyan, K., & Zisserman, A. (2015). Spatial transformer networks. In Advances in Neural Information Processing Systems (pp. 2017-2025).
>
> >> Some of the masks shown in Fig 3 seem rather suspicious and prompt this referee to think that the networks are seriously overfitting to the data.
>
> The train loss vs validation loss difference does not suggest that the proposed model suffers from greater overfitting than the original architecture. Moreover, inspired by this comment, we have designed a test on cluttered MNIST showing that the attention augmented model generalizes better on the test set when increasing the number of distractors (unseen during training), see Figure 4e in the new version of the paper. We hypothesize that attention prevents the model from memorizing uninformative parts of the image, which could be associated with noise. Section 4.1 and the conclusion now reflect this new finding.

---

> ### Author Response · Authors · 2018-01-01
> **Re: AnonReviewer3**
>
> We thank the reviewer for the feedback,
>
> >> First, some claimed are made about how the proposed approach "enhances most of the desirable properties from previous approaches” (see pp 1-2) but these claims are never backed up.
>
> With this sentence we tried to convey that the proposed model accumulates the best of the following properties in the literature: (i) it works in a single pass because it uses a single feed-forward CNN, differently from recurrent and two-step models, (ii) it is trained with SGD instead of RL, thus it presents faster convergence and it does not require sampling, (iii) it can be used to augment any architecture, as we show for WRNs, and (iv) it is simple to implement (instead of creating a whole new network architecture, we just add the attention heads and the attention outputs to an already existing one, eq 9). In order to better back up these properties, we have added a table in the introduction (Table 2) comparing the different architectures in the literature with ours, showing that ours accumulates the best of them.
>
> >> More generally since the paper focuses on attention, other attentional approaches should be used as benchmarks beyond the WRN baseline.
>
> Please note that we include other attentional approaches for fine-grained recognition in all tables:
>   * Table 3 -> FAM [A];
>   * Table 4 -> RA-CNN [B], STN [C], B-CNN [D], PD [E], FCAN [F];
>   * Table 5 -> DVAN [G], FCAN [F], B-CNN [C], RA-CNN [B];
>   * Table 6 -> DVAN [G], FCAN [F], RA-CNN [B].
>
> Moreover, most of these approaches propose singular architectures that have been especially engineered for solving their respective recognition tasks, while the purpose of our approach is to demonstrate that our proposed mechanism works on general purpose architectures.
>
> [A] Rodríguez, P., Cucurull, G., Gonfaus, J. M., Roca, F. X., & Gonzalez, J. (2017). Age and gender recognition in the wild with deep attention. Pattern Recognition, 72, 563-571.
> [B] Fu, J., Zheng, H., & Mei, T. (2017, July). Look closer to see better: recurrent attention convolutional neural network for fine-grained image recognition. In Conf. on Computer Vision and Pattern Recognition.
> [C] Jaderberg, M., Simonyan, K., & Zisserman, A. (2015). Spatial transformer networks. In Advances in Neural Information Processing Systems (pp. 2017-2025).
> [D] Lin, T. Y., RoyChowdhury, A., & Maji, S. (2015). Bilinear cnn models for fine-grained visual recognition. In Proceedings of the IEEE International Conference on Computer Vision (pp. 1449-1457).
> [E] Zhang, N., Donahue, J., Girshick, R., & Darrell, T. (2014, September). Part-based R-CNNs for fine-grained category detection. In European conference on computer vision (pp. 834-849). Springer, Cham.
> [F] Liu, X., Xia, T., Wang, J., & Lin, Y. (2016). Fully convolutional attention localization networks: Efficient attention localization for fine-grained recognition. arXiv preprint arXiv:1603.06765.
> [G] Zhao, B., Wu, X., Feng, J., Peng, Q., & Yan, S. (2016). Diversified visual attention networks for fine-grained object classification. arXiv preprint arXiv:1606.08572.

---

> > ### Comment · AnonReviewer3 · 2018-01-12
> > **Thanks but not enough...**
> >
> > Unfortunately, I do not think this rebuttal addresses my main complaint. I understand that the benchmarks include systems that use attentional mechanisms. My main issue is that the paper is about attention but different attentional mechanisms are never compared on a level play-field (i.e., using the same architectures, optimizers, etc etc). There is no way from the benchmarks to properly assess how much of the improvement is actually driven by the proposed attentional mechanism as opposed to anything else. This is all the more problematic given how small the improvements are. I would also add that with all that said the proposed mechanism remains relatively incremental with respect to related work (work properly cited) and that it seems to be better suited for a more specialized conference.

---

> > > ### Author Response · Authors · 2018-01-12
> > > **Re: Thanks but not enough...**
> > >
> > > Thank you for the thorough review. We think the comments help to keep a high standards on this conference, and our paper has greately improved the quality thanks to them.
> > >
> > > >> Unfortunately, I do not think this rebuttal addresses my main complaint. I understand that the benchmarks include systems that use attentional mechanisms. My main issue is that the paper is about attention but different attentional mechanisms are never compared on a level play-field (i.e., using the same architectures, optimizers, etc etc). There is no way from the benchmarks to properly assess how much of the improvement is actually driven by the proposed attentional mechanism as opposed to anything else.
> > >
> > > We understand the concern, this is exactly why we worked hard during the review period to find the time to include a comparison between STNs and the proposed attention mechanism under the same exact settings (same base architecture, learning algorithm, hyperparameters, training steps, etc.) showing that ours generalizes much better. Moreover, through all the manuscript, we emphasize that our approach is simpler and faster than other competing approaches.
> > >
> > > >> This is all the more problematic given how small the improvements are.
> > >
> > > In the second point of the responses to AnonReviewer3 (3/3) (https://openreview.net/forum?id=rJe7FW-Cb&noteId=BJZ7a4ING&noteId=HkabPkOQM), we explain that the improvement is not so modest given the current context. In our case, improvement is comparable to that found in STN, for example.
> > >
> > > >> There is no way from the benchmarks to properly assess how much of the improvement is actually driven by the proposed attentional mechanism as opposed to anything else.
> > >
> > > We think it is clear that plugging the proposed mechanism into a state-of-the-art CNN results in an improvement. In fact, in every table we show how the augmented models are always better.
> > >
> > > >> I would also add that with all that said the proposed mechanism remains relatively incremental with respect to related work (work properly cited) and that it seems to be better suited for a more specialized conference.
> > >
> > > We still think our work helps indeed to build better representations, and it could be of inspiration for future work in any other field.

---

### Official Review · AnonReviewer2 · 2017-11-28
**Improvement gain is small**

**Rating:** 5
**Confidence:** 4

**Review:**

This paper proposes a feed-forward attention mechanism for fine-grained image classification. It is modular and can be added to any convolutional layer, the attention model uses CNN feature activations to find the most informative parts then combine with the original feature map for the final prediction. Experiments show that wide residual net together with this new attention mechanism achieve slightly better performance on several fine-grained image classification tasks.

Strength of this work:
1) It is end-to-end trainable and doesn't require multiple stages, prediction can be done in single feedforward pass.
2) Easy to train and doesn't increase the model size a lot.

Weakness:
1) Both attention depth and attention width are small. The choice of which layer to add this module is unclear to me.
2) No analysis on using the extra regularization loss actually helps.
3) My main concern is the improvement gain is very small. In Table3, the gain of using the gate module is only 0.1%. It argues that this attention module can be added to any layer but experiments show only 1 layer and 1 attention map already achieve most of the improvement. From Table 4 to Table 7, WRNA compared to WRN only improve ~1% on average.

---

> ### Author Response · Authors · 2018-01-01
> **Re: AnonReviewer2**
>
> Thanks for the feedback,
>
> >> 1) Both attention depth and attention width are small.
>
> Although higher AD and AW do result in an increment of accuracy, we considered that 2 was enough to demonstrate that the proposed mechanism enhances the baseline models at negligible computational cost. In order to address this concern, we have included experiments on deformable mnist where it can be seen that the performance increases with higher AW, and AD (Figure 4b and 4c in the new version of the paper).
>
> >> The choice of which layer to add this module is unclear to me.
>
> Please note that the same placing problem is present in most of the well-known CNN layers such as Dropout, Local-contrast normalization, Spatial Transformers, etc.
> However, as we answer to R1’s first question, we have established a systematic methodology which consists in adding the attention mechanism after each subsampling layer of the WRN in order to obtain features of different levels at the smallest possible computational cost. This information is now included at the end of section 3.3, when Table 2 is introduced, and in the second paragraph of section 4.
>
> >> 2) No analysis on using the extra regularization loss actually helps.
>
> The analysis has been included in section 4.1, where experiments with Cluttered Translated MNIST show that regularization adds an extra performance increment.
>
> >> 3) My main concern is the improvement gain is very small. In Table3, the gain of using the gate module is only 0.1%. It argues that this attention module can be added to any layer but experiments show only 1 layer and 1 attention map already achieve most of the improvement.
>
> We hope that the new experiments on Cluttered Translated MNIST in section 4.1 help to clarify this point. Also, as it can be seen in Figure 4d, gates are crucial when AD and AW grow.
>
> >> From Table 4 to Table 7, WRNA compared to WRN only improve ~1% on average.
>
> Please note that 1% is a remarkable amount given that, for instance, other relevant papers such as Spatial Transformer Networks only reported an improvement of 0.8% on CUB200-2011 with respect to their own baseline (see table 3 in their work). Moreover, in the case of residual attention networks [B], the reported improvement on Cifar100 is 0.05%.
>
> [A] Jaderberg, M., Simonyan, K., & Zisserman, A. (2015). Spatial transformer networks. In Advances in Neural Information Processing Systems (pp. 2017-2025).
> [B] Wang, F., Jiang, M., Qian, C., Yang, S., Li, C., Zhang, H., ... & Tang, X. (2017). Residual Attention Network for Image Classification. CVPR2017.

---

### Official Review · AnonReviewer1 · 2017-12-01
**Review for A Painless Attention Mechanism for Convolutional Neural Networks**

**Rating:** 6
**Confidence:** 4

**Review:**

Paper presents an interesting attention mechanism for fine-grained image classification. Introduction states that the method is simple and easy to understand. However, the presentation of the method is bit harder to follow. It is not clear to me if the attention modules are applied over all  pooling layers. How they are combined?

Why use cross -correlation as the regulariser? Why not much stronger constraint such as orthogonality over elements of M in equation 1? What is the impact of this regularisation?

Why use soft-max in equation 1? One may use a Sigmoid as well? Is it better to use soft-max?

Equation 9 is not entirely clear to me. Undefined notations.

In Table 2, why stop from AD= 2 and AW=2?  What is the performance of AD=1, AW=1 with G? Why not perform this experiment over all 5 datasets? Is this performances, dataset specific?

The method is compared against 5 datasets. Obtained results are quite good.

---

> ### Author Response · Authors · 2018-01-01
> **Re: AnonReviewer1**
>
> Thank you for your comments,
>
> >> However, the presentation of the method is bit harder to follow. It is not clear to me if the attention modules are applied over all  pooling layers.
>
> Any layer of the network can be augmented with the attention mechanism. We chose to use the augmentation after each pooling layer in order to reduce even further the computational cost. We have clarified this point at the end of section 3.3, when Table 2 is introduced, and in the second paragraph of section 4.
>
> >> How they are combined?
>
> As it can be seen in Fig 1, 2a, and 2b, a 1x1 convolution is applied to the output of the layer we want to augment, producing an attentional heatmap. This heatmap is then element-wise multiplied with a copy of the layer output, and the result is used to predict the class probabilities and a confidence score. This process is applied to an arbitrary number N of layers, producing N class probability vectors, and N confidence scores. Then all the class predictions are weighted by the confidence scores (softmax normalized so that they add-up to 1) and averaged (using Eq 9). This is the final combined prediction of the network. This overall explanation is now placed in the “Overview” section before section 3.1.
>
> >> Why use cross -correlation as the regulariser? Why not much stronger constraint such as orthogonality over elements of M in equation 1?
>
> Please note that the 2-norm operation requires to square all the elements of the matrix, thus the minimum norm is achieved when the inner product of all the different pairs of masks is 0 (orthogonal). Thus, orthogonality is constrained by regularizing the 2-norm of a matrix. This is now clarified after Eq 3.
>
> >> What is the impact of this regularisation?
>
> In order to address questions R1.1, etc.. we have added experiments on deformable mnist, showing the importance of each module. In figure 4d it can be seen that the regularized model performs better than the unregularized counterpart.
>
> >> Why use soft-max in equation 1? One may use a Sigmoid as well? Is it better to use soft-max?
>
> We use softmax because it constrains the network to choose only one region in the image, thus forcing it to learn which is the most discriminative region. Using sigmoids attains the risk of just learning to predict 1s for every region, or all zeros. Note that multiple regions can still be identified by using multiple attention heads. This explanation has been included in section 3.1.
>
> >> Equation 9 is not entirely clear to me. Undefined notations.
>
> “output” is the predicted vector of class probabilities, “g_net” is the confidence score for the original output of the network “output_net” (without attention). This information has been appended after equation 9.
>
> >> In Table 2, why stop from AD= 2 and AW=2?  What is the performance of AD=1, AW=1 with G? Why not perform this experiment over all 5 datasets?
>
> We had to constrain the number of experiments to a limited amount of time and resources, which makes it difficult to brute-force all hyperparameter combinations with all datasets. We hope that this question is now clarified with the experiments on deformable-mnist (Section 4.1 in the new version of the paper).
>
> >> Is this performances, dataset specific?
>
> No, generally increasing AD and AW results in better performance in all datasets.

---

### Author Response · Authors · 2018-01-01
**Paper v2.0**

We thank all the reviewers for their highly valuable feedback. We have addressed all comments one by one, and we have accordingly updated the manuscript. Changes appear in blue.
List of changes:
  * A new table (Table 2) in the introduction has been added to clarify the advantages of our proposal with respect to the literature.
  * An overview section has been added to section 3 (Section 3.1) to summarize and clarify how the different submodules fit together.
  * Undefined notations have been clarified in equation 9.
  * Ablation experiments on cluttered translated MNIST have been introduced in section 4.1.
  * Textual clarifications addressing comments from the reviewers.

Thanks to these improvements resulting from the review process, the manuscript has substantially clarified the contribution of our work, has improved the technical quality and has also enhanced the experimental quality. We trust these improvements make it even more appealing for publication at this conference.

---

### Public Comment · (anonymous) · 2018-01-22
**Relevant paper with spatial regularization and attention**

Just for completeness, a relevant paper that learns spatial regularizations using an attention mechanism on a final ResNet representation is [1].

[1] Zhu, F., Li, H., Ouyang, W., Yu, N., & Wang, X. Learning Spatial Regularization with Image-level Supervisions for Multi-label Image Classification. in CVPR 2017

---

> ### Author Response · Authors · 2018-01-22
> **Re: Relevant paper with spatial regularization and attention**
>
> Thank you for the interest in our paper. We have included this explanation in the “Related Work” section, as a specialized solution for multilabel classification, where instead of learning universal modules, a ResNet is modified to improve its multilabel classification by enhancing the predictions with the learned most relevant regions.
>
> Differently from this ICLR, in [1], instead of designing a general mechanism like the one proposed in our submission, the authors design an specialized attention mechanism for multilabel classification and test it on MSCOCO, NUS-WIDE, and WIDER. Namely, they use the features in “res4b22 relu” in order to extract attention scores for each label through three convolutional layers. To avoid attending to labels not present for the input being processed, these attention maps are multiplied by “confidence maps”, which are learned to be 1 if the label is present, and 0 if not. The attentional predictions are average with the network predictions. Differently, we want to incorporate fine detail at different levels of abstraction to the final prediction, thus, we propose a general “Attention Module”, that can be applied at many levels to any network, to enhance the final prediction weighted by the relevance of each prediction (for instance, details in the texture might help do distinguish between two birds which are similar at abstract level).
>
> Changes will be visible in the final version of the paper.

---

### Author Response · Authors · 2018-10-25
**Article Update**

A new revised version of this paper has been published at ECCV2018.

Rodríguez, P., Gonfaus, J. M., Cucurull, G., Roca, F. X., & Gonzàlez, J. (2018, September). Attend and Rectify: A Gated Attention Mechanism for Fine-Grained Recovery. In European Conference on Computer Vision (pp. 357-372). Springer, Cham.

http://openaccess.thecvf.com/content_ECCV_2018/html/Pau_Rodriguez_Lopez_Attend_and_Rectify_ECCV_2018_paper.html

---

### Decision · Program_Chairs · 2018-01-29
**ICLR 2018 Conference Acceptance Decision**

**Decision:**

Reject

**Comment:**

This paper received borderline reviews. Initially, all reviewers raised a number of concerns (clarity, small improvements, etc). Even after some back and forth discussion, concerns remain, and it's clear that while the idea has potential, another round of reviewing is needed before a decision can be reached. This would be a major revision in a journal. Unfortunately, that is not possible in a conference setting and we must recommend rejection. We recommend the authors to use the feedback to make the manuscript stronger and submit to a future venue.